# A Chronic Disease in Adolescence and Selection to an Educational Path—A Longitudinal Study

**DOI:** 10.3390/ijerph192114407

**Published:** 2022-11-03

**Authors:** Leena Koivusilta, Riittakerttu Kaltiala, Anna Myöhänen, Risto Hotulainen, Arja Rimpelä

**Affiliations:** 1Department of Social Research, Faculty of Social Sciences, University of Turku, 20014 Turku, Finland; 2Unit of Health Sciences, Faculty of Social Sciences, Tampere University, 33014 Tampere, Finland; 3Faculty of Medicine and Health Technology, Tampere University, 33014 Tampere, Finland; 4Department of Adolescent Psychiatry, Tampere University Hospital, 33521 Tampere, Finland; 5Centre for Educational Assessment, Faculty of Educational Sciences, University of Helsinki, 00014 Helsinki, Finland

**Keywords:** school performance, academic path, lower secondary school, asthma, diabetes, epilepsy

## Abstract

Chronic disease may affect adolescents’ educational success. We study whether adolescents with a somatic chronic condition have lower school performance, lower odds for academic education, and a delayed start of upper-secondary studies. Seventh graders and ninth graders in the Helsinki Metropolitan Region, Finland, were invited to participate in a school survey in 2011 and 2014, respectively. The respondents (2011, *N* = 8960; 2014, *N* = 7394) were followed using a national application registry until 2017. The chronic conditions were asthma, diabetes, and epilepsy. Outcomes were grade point average (GPA), study place in an academic school, and delayed start of secondary education. Adolescents with a chronic disease needing medication had lower GPAs in both grades. Chronic disease with medication in the seventh grade predicted higher odds for the non-academic track (OR = 1.3) and the delayed start (OR = 1.4). In the ninth grade, chronic disease predicted non-academic studies univariately (OR = 1.2) and was not associated with the delayed start. The somatic chronic condition with medication, particularly epilepsy, slightly lowers students’ school performance, which is a mediator between the chronic condition and selection into educational paths. Compared to gender and parents’ education, and particularly to GPA, the role of chronic conditions on educational outcomes is small.

## 1. Introduction

Low education is a strong predictor of poor health and early death in adult age [1,2]. In adolescence, poor academic achievement [3] and dropping out of education [4] increase the risks for poor health outcomes. Critical decisions on education taken in adolescence shape the pathways from childhood socioeconomic positions towards one’s own educational and socioeconomic career. Educational resources obtained during education may impact health through various mechanisms. Among these are knowledge and skills which may affect a person’s cognitive functioning and readiness to receive and apply health-related information, as well as an ability to communicate and use health services [5]. Health literacy gained during education has been observed as a potential mechanism through which individuals’ educational resources safeguard their health [6].

In many European countries, students are sorted relatively early (before age 13) into separate tracks, whereas in other countries (e.g., Finland, the other Nordic countries) all students follow mainly the same curriculum through their primary and lower-secondary school [7,8]. While choices for educational paths are available, they may be limited by economic, geographical, or cultural conditions, parents’ education, or individual reasons like poor health. International PISA studies have shown how the socio-economic position of the family shapes a child’s academic performance [9]. Other studies have shown how the material, cultural, and intellectual resources owned by the families influence the children’s educational choices and shape their careers [10]. Children who are not able to use educational opportunities are at risk of experiencing disadvantages over their life course, such as difficulties in entering the labor market or finding an economically rewarding position [11,12].

In the turbulent years of adolescence, with its special developmental tasks, a chronic condition brings an extra challenge for schooling and learning. Over 10% of adolescents have a chronic disease [13], which may disturb their coping with schoolwork, lower their academic engagement, and increase school absenteeism. Students with a chronic condition more often repeat a grade, encounter academic challenges, and have higher school absenteeism compared to those without [14,15]. Students with a chronic condition also have lower educational attainment; they less often achieve high school diplomas or college graduation, and drop out of education more often than their healthy peers [14,16,17,18,19]. Research on educational outcomes has often concentrated on studying the impact of mental health problems., For example, analysis of a register-based follow-up data from a Finnish 1987 birth cohort showed that the probability of the NEET status (not in education, employment, or training) was higher for adolescents who received treatment for psychiatric disorders [20]. However, research on the impact of somatic diseases on educational outcomes is scarce.

The setting for our study is Finland, a Nordic welfare society where educational career choices take place late (at age 16) and where school health service, school welfare groups [21], and three-tiered learning support [22] are available in all schools. Specialist healthcare takes care of most of the children with chronic conditions. After nine years of comprehensive school, students apply for upper-secondary education. They are sorted according to their application preferences and grade point averages (GPAs). The upper-secondary schools are divided into two main lines: academic (general upper-secondary) and vocational tracks. Those who are unsure about their study choice, can continue in the 10th grade to refine their further study plans, and improve the grades in their graduation report. Even though there are alternative routes, those selected to vocational schools have higher odds for lower education later in life [23,24].

We study here if adolescents with serious somatic chronic conditions have lower GPAs in their graduation reports, lower odds for academic education, and delayed starts of upper-secondary studies.

## 2. Materials and Methods

### 2.1. Study Design and Participants

The learning and health of students from the Helsinki Metropolitan Region were surveyed in the seventh grade (12–13-year-olds, 2011) and in the ninth grade (15–16-year-olds, 2014) (MetLoFin study). All comprehensive schools in the region with seventh and ninth grades were invited, thus constructing a total sample of the students in the region. In this study, participants from special schools (2011, *N* = 4; 2014, *N* = 16) were excluded.

The protocol was approved by the Ethical Committee of the Finnish Institute for Health and Welfare. Parental consent was obtained in two of 14 municipalities where local authorities required it. Information letters were sent to parents in other municipalities. The online surveys were conducted as a part of normal schoolwork. Participation was voluntary. Students were instructed that they could decline to answer any question or withdraw from the survey at any time [25].

Registry data on students’ applications for upper-secondary schools were obtained from the Finnish National Agency for Education. This is a national registry covering all upper-secondary schools in Finland. In practice, all students apply via the Joint Application System when completing the ninth grade. The selection is based on school marks from the graduation report and students’ preferences. There are two general application rounds each year followed by additional rounds where students can apply for vacant places. The applications were followed from spring 2014 (graduation time) to spring 2017. The survey answers and the joint application system data were merged. Of 13,012 students in 2011, 8960 (69%) from 127 schools answered the questionnaire and had application data available. In 2014, the corresponding numbers were 7394 of 13,138 (56%) and 124.

### 2.2. Outcome Variables

Three outcome variables were used: Grade point average (GPA), Non-academic track, and Delayed start of upper-secondary school. GPA in the graduation report (end of the ninth grade) was computed as the mean of school marks for foreign language, mother tongue, math, and science (mean of physics, chemistry, biology, geography) obtained from the joint application system. In Finland, 4 is a fail mark and 10 is the best mark possible. GPA was used as a decimal number or categorized (high = 9–10, middle = 7–8.99, low = 4–6.99).

Non-academic track consisted of students who were selected to vocational schools (2011, *N* = 3134; 2014, *N* = 2444) and those who had no study place according to the registry (2011, *N* = 264; 2014, *N* = 200). The last group was placed here because they were likely selected to vocational schools for open places after the application period, but information had not been reported to the registry. The final variable was dichotomous: academic vs. non-academic track. The latest accepted application was used to place the student. Some students had participated in the survey of the same cohorts in 2016 [26]. If a student was found in a different track than the registry placement showed and had not reapplied after 2016, the participant’s placement was revised (2011, *N* = 91; 2014, *N* = 85).

Delayed start. Some students applied several times because they were not accepted, had not got a desired place, or had interrupted. Those who did not continue studies directly after graduating had a delayed start.

### 2.3. Explanatory Variables and Covariates

Chronic disease. We selected somatic diseases likely to disturb schoolwork (asthma, diabetes, and epilepsy), using earlier literature and medical knowledge based on students’ self-reports to the question: “Do you have a chronic disease or disability”. In addition, the following diseases were enquired about: asthma, musculoskeletal condition, diabetes, allergic rhinitis, hay fever or other allergy (separated in 2014), epilepsy, mental health problem, other. Students were further asked if they used regularly or almost regularly prescribed medication and for which disease: asthma, diabetes, allergic rhinitis or hay fever, other allergy, epilepsy, mental health problem, pain, and aches, and other. Students could tick several options in both questions. The final variable was categorized: no chronic disease, chronic disease without medication, and chronic disease with medication.

Cross-tabulations of the above questions showed some inconsistencies and implausible answers. We removed respondents (2011, *N* = 1; 2014, *N* = 109) who reported an unconvincing number of diseases/medicines (≥5 in 2011; ≥6 in 2014). Most of those had ticked all options. Second, we checked open answers to the options “other disease” and “other medicine”. We excluded participants with inappropriate and improper answers (e.g., sexual-related matters, YouTube links, joking, mickey-taking). Finally, we checked case by case those who reported epilepsy or diabetes. We used open text and the question on harm experience due to a disease (this question could not be used otherwise, because it did not separate between diseases) and excluded those with implausible combinations. Altogether 20 cases from 2011 and 168 cases from 2014 data were removed. The final variable was classified: chronic disease without medicine, chronic disease with medicine, no chronic disease. The diseases in the variable were asthma, diabetes, and epilepsy.

Parents’ education. Parents’ education was dichotomous “high” and “middle/low”. Matriculation examination and polytechnics or university degrees were coded high. If a participant reported “No mother and father”, the answer was coded as missing. Appendix C shows how the chronic diseases were distributed by gender and parents’ education.

### 2.4. Statistical Methods

Linear regression analyses were at first used to construct the models of the impact of chronic disease, gender, and parents’ education on the first outcome: grade point average (GPA). Next, binary logistic regression analyses were performed for the second and third outcomes: ending up to non-academic secondary school and delayed start, including the same covariates but now also GPA as an explanatory variable. Odds ratios (OR) and their 95% confidence intervals were computed. Because of potential comorbidities of mental health problems and somatic chronic diseases, an adjustment for self-reported mental health problems was performed in all models.

The corresponding tables, as performed by logistic models, are presented by average marginal effects analysis in Appendix D. There was no difference in Table 1 and Table 2. In Table 3 disease with medication was not significant in Model 1, but was significant in Model 2 like in the odds ratio analyses. No differences were seen in Table 4 and Table 5. 

All statistical analyses were performed by IBM SPSS Statistics for Windows (Version 28.0. Armonk, NY, USA: IBM Corp, released 2021) except average marginal effects of the Appendix D were computed by margins library of R.

## 3. Results

In the seventh grade, 8.4% of students (*N* = 753; *N* = 332 girls; *N* = 421 boys) reported having a chronic disease while in the ninth grade, the corresponding figure was 9.6% (N = 708; *N* = 309 girls; *N* = 399 boys).

Students who had a disease with medication in either one of the grades had lower GPAs than those who did not have the disease or had the disease but without medication (Table 1). The association persisted when the covariates (gender, parents’ education) were added in the model (Model 2) and also when the variable indicating a mental health problem was added (Model 3). Boys had lower GPAs compared to girls, and students whose parents had high education or no mental health problems had higher GPA.

**Table 1 ijerph-19-14407-t001:** The association of chronic disease in the seventh grade (*N* = 8960) and in the ninth grade (N = 7394) with the grade point average (GPA ^a^) in bivariate (Model 1) and adjusted models (Models 2 and 3). Linear regression analyses.

Seventh Grade						
Explanatory Variable	Model 1 ^b^		Model 2 ^c^		Model 3 ^d^	
	B (SE)	*p* Value	B (SE)	*p* Value	B (SE)	*p* Value
Chronic disease						
Disease without medication (=yes)	−0.13 (0.08)	0.10	−0.09 (0.08)	0.21	−0.09 (0.08)	0.240
Disease with medication (=yes)	**−0.17 (0.05)**	**<0.001**	**−0.14 (0.04)**	**0.001**	**−0.14 (0.04)**	**<0.001**
Gender (=boy)	**−0.47 (0.02)**	**<0.001**	**−0.46 (0.02)**	**<0.001**	**−0.46 (0.02)**	**<0.001**
Parents’ education (=high)	**0.77 (0.02)**	**<0.001**	**0.77 (0.02)**	**<0.001**	**0.77 (0.02)**	**<0.001**
Mental health problem (=yes)	**−0.40 (0.13)**	**0.002**			**−0.49 (0.12)**	**<0.001**
**Ninth Grade**						
	**Model 1 ^b^**		**Model 2 ^c^**		**Model 3 ^d^**	
	**B (SE)**	***p* Value**	**B (SE)**	***p* Value**	**B (SE)**	***p* Value**
Chronic disease						
Disease without medication (=yes)	−0.07 (0.08)	0.42	−0.03 (0.08)	0.70	−0.03 (0.08)	0.736
Disease with medication (=yes)	**−0.18 (0.05)**	**<0.001**	**−0.14 (0.05)**	**0.003**	**−0.14 (0.05)**	**0.003**
Gender (=boy)	**−0.44 (0.02)**	**<0.001**	**−0.45 (0.02)**	**<0.001**	**−0.46 (0.02)**	**<0.001**
Parents’ education (=high)	**0.74 (0.03)**	**<0.001**	**0.76 (0.03)**	**<0.001**	**0.75 (0.03)**	**<0.001**
Mental health problem (=yes)	**−0.16 (0.08)**	**0.04**			**−0.23 (0.07)**	**0.002**

^a^ GPA is based on the final school marks from lower secondary school. ^b^ Chronic disease, gender, parents’ education and mental health problem each in a separate analysis. ^c^ Adjusted for parents’ education and gender. ^d^ Adjusted for parents’ education, gender, and mental health problem. The statistically significant associations are marked in bold.

Students who had a disease with medication in the seventh grade had higher odds of the non-academic track (Table 2, Model 1). When gender and parents’ education were added to the model, the disease variable maintained its significance (Model 2). GPA was the most powerful predictor, and the inclusion of it in the model caused the vanishing of the association (Model 3). The associations did not change when the mental health variable was added (Model 4).

**Table 2 ijerph-19-14407-t002:** The association of chronic disease in the seventh grade (*N* = 8960) up to the non-academic upper-secondary school. Bivariate (Model 1) and adjusted logistic regression models (Models 2–4). Odds ratios (OR) and their 95% confidence intervals.

	Seventh Grade			
Explanatory Variable	Model 1 ^a^	Model 2 ^b^	Model 3 ^c^	Model 4 ^d^
	OR (95% CI)	OR (95% CI)	OR (95% CI)	OR (95% CI)
Chronic disease				
No	1.0	1.0	1.0	1.0
Disease without medication	1.2 (0.9 -1.6)	1.1 (0.8–1.5)	1.1 (0.7–1.6)	1.1 (0.7–1.6)
Disease with medication	**1.3 (1.1–1.5)**	**1.3 (1.05–1.5)**	1.1 (0.9–1.4)	1.1 (0.9–1.4)
Gender				
Girl	1.0	1.0	1.0	1.0
Boy	**1.7 (1.6–1.8)**	**1.8 (1.6–1.9)**	**1.2 (1.0–1.3)**	**1.2 (1.0–1.3)**
Parents’ education				
High	1.0	1.0	1.0	1.0
Middle/low	**3.9 (3.6–4.3)**	**4.0 (3.6–4.4)**	**2.5 (2.3–2.8)**	**2.5 (2.3–2.8)**
Grade point average				
High	1.0		1.0	1.0
Middle	**12.6 (9.8–16.2)**		**10.5 (8.1–13.5)**	**10.4 (8.1–13.4)**
Low	**346 (256–468)**		**252 (186–342)**	**251 (185–340)**
Mental health problem				
No	1.0			1.0
Yes	**2.1 (1.4–3.4)**			**2.1 (1.2–3.6)**

^a^ Bivariate model. Each explanatory variable was analyzed in a separate analysis. ^b^ Chronic disease, gender, and parents’ education as explanatory variables in the model. ^c^ Adjusted for gender, parents’ education, and GPA. ^d^ Adjusted for gender, parents’ education, GPA, and mental health problem. The statistically significant associations are marked in bold.

In the ninth grade, disease with medication was associated with ending up to non-academic track univariately but not in the adjusted models (Table 3). Low GPA was a powerful predictor of the non-academic track in all models. Also male gender, parents’ low education, and mental health problem predicted the non-academic track in all models.

**Table 3 ijerph-19-14407-t003:** The association of chronic disease in the ninth grade (*N* = 7394) with ending up to the non-academic upper-secondary school. Bivariate (Model 1) and adjusted logistic regression models (Models 2–4). Odds ratios (OR) and their 95% confidence intervals.

	Ninth Grade			
Explanatory Variable	Model 1 ^a^	Model 2 ^b^	Model 3 ^c^	Model 4 ^d^
	OR (95% CI)	OR (95% CI)	OR (95% CI)	OR (95% CI)
Chronic disease				
No	1.0	1.0	1.0	1.0
Disease without medication	1.1 (0.8–1.4)	1.0 (0.8–1.4)	1.0 (0.7–1.5)	1.0 (0.7–1.4)
Disease with medication	**1.2** **(1.04–1.5)**	1.2 (1.0–1.4)	1.1 (0.8–1.3)	1.0 (0.8–1.3)
Gender				
Girl	1.0	1.0	1.0	1.0
Boy	**1.7 (1.5–1.8)**	**1.8 (1.6–2.0)**	**1.2 (1.1–1.3)**	**1.2 (1.1–1.4)**
Parents’ education				
High	1.0	1.0	1.0	1.0
Middle/low	**3.9 (3.5–4.4)**	**4.1 (3.6–4.6)**	**2.6 (2.2–3.0)**	**2.6 (2.2–3.0)**
Grade point average				
High	1.0		1.0	1.0
Middle	**15.2 (11.2–20.5)**		**13.3 (9.8–18.0)**	**13.3 (9.8–18.0)**
Low	**346 (244–489)**		**273 (193–387)**	**273 (192–387)**
Mental health problem				
No	1.0			1.0
Yes	**1.6 (1.2–2.1)**			**1.7 (1.2–2.4)**

^a^ Bivariate model. Each explanatory variable was analyzed in a separate analysis. ^b^ Chronic disease, gender, and parents’ education as explanatory variables in the model. ^c^ Adjusted for gender, parents’ education, and GPA. ^d^ Adjusted for gender, parents’ education, GPA, and mental health problem. The statistically significant associations are marked in bold.

The disease with medication in the seventh grade was also significantly associated with the delayed start of upper-secondary education when adjusted for gender and parents’ education, but was not associated after adjusting for GPA (Table 4). GPA was the most significant predictor and when added in the model (Model 3), boys’ probability to start studies late was smaller compared to that of girls. Parents’ education was a significant predictor in all models. Adjustment for a mental health problem did not change the associations (Model 4).

**Table 4 ijerph-19-14407-t004:** The association of chronic disease in the seventh grade (*N* = 8938) with the delayed start of upper-secondary school. Bivariate (Model 1) and adjusted logistic regression models (Model 2–4). Odds ratios (OR) and their 95% confidence intervals.

	Seventh Grade			
Explanatory Variable	Model 1 ^a^	Model 2 ^b^	Model 3 ^c^	Model 4 ^d^
	OR (95% CI)	OR (95% CI)	OR (95% CI)	OR (95% CI)
Chronic disease				
No	1.0	1.0	1.0	1.0
Disease without medication	0.8 (0.4–1.6)	0.8 (0.4–1.6)	0.8 (0.4–1.5)	0.7 (0.4–1.5)
Disease with medication	**1.4 (1.0–1.9)**	**1.4 (1.0–2.0)**	1.3 (1.0–1.8)	1.3 (1.0–1.8)
Gender				
Girl	1.0	1.0	1.0	1.0
Boy	1.0 (0.8–1.2)	1.0 (0.8–1.1)	**0.7 (0.6–0.9)**	0.7 (0.6–0.9)
Parents’ education				
High	1.0	**1.0**	1.0	1.0
Middle/low	**2.0 (1.7–2.4)**	**2.0 (1.7–2.4)**	**1.3 (1.1–1.5)**	**1.3 (1.1–1.5)**
Grade point average				
High	1.0		1.0	1.0
Middle	**4.4 (2.9–6.9)**		**4.4 (2.8–6.9)**	**4.4 (2.8–6.9)**
Low	**14.6 (9.4–22.7)**		**14.4 (9.1–22.6)**	**14.3 (9.1–22.4)**
Mental health problem				
No	1.0			1.0
Yes	**2.1 (1.1–4.3)**			**1.8 (0.9–3.7)**

^a^ Bivariate model. Each explanatory variable was analyzed in a separate analysis. ^b^ Chronic disease, gender, and parents’ education as explanatory variables in the model. ^c^ Adjusted for gender, parents’ education, and GPA. ^d^ Adjusted for gender, parents’ education, GPA, and mental health problem. The statistically significant associations are marked in bold.

In the ninth grade, the disease was not associated with delayed start, and gender was associated only in Model 3 (Table 5). GPA was significant in both the bivariate model and in Model 3. Parents’ education was statistically significant in the bivariate analysis (Model 1) and in Model 2, but not when GPA was added (Model 3). Adding the variable of a mental health problem did not change the associations (Model 4).

We conducted sensitivity analyses, in which we had each disease separately in the regression models (Appendix A and Appendix B), the association of each disease with the outcome variables was of similar direction as that of the combined chronic disease variable. Associations with both GPA and ending up to a non-academic upper-secondary school were stronger for epilepsy than for the two other diseases.

**Table 5 ijerph-19-14407-t005:** The association of chronic disease in the ninth grade (*N* = 7384) with the delayed start of upper-secondary school. Bivariate (Model 1) and adjusted logistic regression models (Model 2– 4). Odds ratios (OR) and their 95% confidence intervals.

		Ninth Grade		
Explanatory Variable	Model 1 ^a^	Model 2 ^b^	Model 3 ^c^	Model 4 ^d^
	OR (95% CI)	OR (95% CI)	OR (95% CI)	OR (95% CI)
Chronic disease				
No	1.0	1.0	1.0	1.0
Disease without medication	0.8 (0.4–1.6)	0.8 (0.4–1.6)	0.8 (0.4–1.6)	0.8 (0.4–1.6)
Disease with medication	1.3 (0.9–1.8)	1.2 (0.9–1.8)	1.2 (0.8–1.7)	1.2 (0.8–1.7)
Gender				
Girl	1.0	1.0	1.0	1.0
Boy	1.1 (0.9–1.3)	1.1 (0.9–1.3)	**0.8 (0.7–1.0)**	0.8 (0.7–1.0)
Parents’ education				
High	1.0	1.0	1.0	1.0
Middle/low	**1.7 (1.4–2.1)**	**1.7 (1.4–2.1)**	1.0 (0.8–1.3)	1.0 (0.8–1.3)
Grade point average				
High	1.0		1.0	1.0
Middle	**5.1 (3.0–8.6)**		**5.2 (3.1–8.9)**	**5.2 (3.1–8.8)**
Low	**17.8 (10.5–30.2)**		**18.6 (10.8–32.0)**	**18.4 (10.7–31.6)**
Mental health problem				
No	1.0			1.0
Yes	**1.9 (1.2–3.1)**			**1.7 (1.0–2.8)**

^a^ Bivariate model. Each explanatory variable was analyzed in a separate analysis. ^b^ Chronic disease, gender and parents’ education as explanatory variables in the model. ^c^ Adjusted for gender, parents’ education and GPA. ^d^ Adjusted for gender, parents’ education, GPA and mental health problem. The statistically significant associations are marked in bold.

## 4. Discussion

School performance (GPA) was slightly lower among students with chronic diseases needing medication compared to those who did not have the disease or whose disease did not require medication. Those who used medication had slightly higher odds of ending up to the non-academic track, but this association disappeared when adjusted for GPA in the graduation report. The delayed start of upper-secondary school was associated with chronic disease in the seventh grade but not in the ninth grade. Low GPA was the most powerful predictor in the models. Male gender, low parents’ education, and mental health problems predicted all three outcomes. The associations were stronger for epilepsy than for diabetes or asthma.

Our findings support earlier studies where chronic conditions [14,16,17,18,27] or special needs [16] were associated with indicators of poorer educational attainment. Chronic health conditions studied mostly comprise asthma, epilepsy, cancer, juvenile arthritis, kidney disease, diabetes, gastrointestinal diseases, or heart conditions in different combinations of diseases [14,15,16,28,29].

The mechanisms through which an illness influences school achievement can be direct or indirect. Knight and Perfect (2019) [30] demonstrated a direct effect in their study; diabetic adolescents whose glucose levels were frequently out of the target range had a higher risk for performing below their true academic potential. An Australian study showed that children with type 1 diabetes did not significantly differ from their peers in the studied indicators of school performance, but poorer glycaemic control was associated with a lower test score [31]. Martinez and Ercikan (2009) [32] showed that chronically ill children performed less well in a standard test of mathematical skills and problem solving. Many chronic conditions directly impact neurocognitive functioning with understandable harm for learning and achievement [33,34].

The effect of the chronic condition on educational attainment may also be indirect. Chronically ill adolescents may have lower educational aspirations and expectations, which affect the educational career [18,35], Further, social exclusion, absences from school [12,31], emotional distress, and mental health problems related to chronic conditions [33,34] may be mediating factors. Chronic conditions are known to increase the risk of mental health problems [34]. The reciprocal association between psychological symptoms and negative school experiences often has a negative impact on achievement [23,25,27,36]. People with mental health problems often face discrimination and may be stigmatized—by themselves, as well as other people—and this may have a further negative impact on self-concept and faith to personal abilities [37].

Our associations observed for epilepsy were stronger than for the other diseases. This suggests that adolescents with epilepsy may have severe problems in their educational careers. Childhood and adolescent onset of epilepsy has been found to predict a low socioeconomic position, educational level included [38]. One possible mechanism is that the disease may lead to becoming labelled as a deviant or less capable person. This may reduce resources needed in making educational decisions, and more so, if important adults, e.g., professionals giving career advice, have prejudices about the young persons’ abilities and chances of success [39].

The strong role played by GPA for the selection of the educational track and delayed start was obvious because it is the most important selection criterium for a study place. The vanishing association between the track and the disease at the seventh grade when adjusting for GPA shows how the disease influences academic performance and how GPA works as a mediator between the chronic disease and association. This has been suggested by some other studies as well [15,27].

Our study confirmed earlier findings on boys’ lower school performance compared to girls and the significance of parents’ education for children’s school attainment [9,40]. The attitudes of Finnish students towards reading, especially for boys, have become more negative, according to the PISA study [9]. High-educated parents are more often able to support their children’s cognitive development [41,42], as well as use their material, cultural, and social compensatory resources to promote their children’s learning and educational careers [43].

Chronic illnesses needing medication were a contributing factor in dividing adolescents into different educational tracks, but did not influence the smoothness of the transition from lower- to upper-secondary education. The effects were, however, small. The Finnish advanced system of school health service and student welfare support [21], as well as the means to help pupils with learning difficulties [22], have likely contributed to the small effect. These may have helped students cope with disease and medications at school and better understand the limitations of the disease. Understanding of the role of chronic diseases in students’ learning and attitudes to education is still scarce. Disease-specific studies would illuminate the phenomena more specifically. Further, research from other countries may help to understand the role of health and support services in minimizing the negative influences of the diseases. Parents’ education was used here as a socio-economic indicator of students’ family background. In future research, a wider set of indicators would be useful to describe adolescents’ varying life contexts, e.g.,in the framework of the socioecological model [44].

## 5. Conclusions

Our results showed that in adolescence, chronic conditions that need medication may negatively affect students’ school performance. Despite the quite modest associations, a chronic condition may act as a selection factor in the transition from lower- to upper-secondary education. Further research is needed on how single diseases and their comorbidities affect educational outcomes, and if the influence of chronic diseases varies according to educational system, arrangement of learning support, and quality of adolescent health care. A chronic condition in adolescence, especially one which requires medication, may be one of the health selection mechanisms causing health inequality in adulthood [45]. Improving the learning and quality of school welfare services is likely to help students with a chronic condition achieve their full academic potential.

## Data Availability

The data presented in this study may be available upon a well-argued, detailed research plan from a non-profit university or research institution on request from Professor Arja Rimpelä, Tampere University and Professor Risto Hotulainen, University of Helsinki. All metadata are in Finnish.

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
