# Peer review of "A Chronic Disease in Adolescence and Selection to an Educational Path—A Longitudinal Study"

_ijerph, 2022, doi:10.3390/ijerph192114407_

Round 1

Reviewer 1 Report

General comments

=============

This paper aims to discuss weather adolescents with a somatic chronic condition have lower school performance? I would like to express several concerns and provide some comments and suggestions as follows. Hopefully, the following comments and suggestions will be helpful for improving this paper.

=============

Major comments

---------------------

1. This paper still lacks further explanation of the learning theory, especially the reference to the past research about the low education is a strong predictor of poor health and early death in adult age.

2. For the literature, it is recommended that the literature related to compare studies from other countries, in particular the Finnish database for this study, and to indicate the results of studies in other countries.  

Minor comments

---------------------

4. This study used the s Linear regression analyses method. And 7th grade (N=8964) and 9th grade (N=7410) with 161 ending up to the non-academic upper secondary school. Is there any difference in the sample compared to other research? The other hand, Do the other control variables influence the result? Ex: family income?

5.The authors can discuss results, implications for managers and scope for future papers in different sections to enhance readability

This article still needs further explanation of the data, and the sample is not sufficient, it is recommended to make minor revise.

Author Response

Review 1

This paper aims to discuss weather adolescents with a somatic chronic condition have lower school performance? I would like to express several concerns and provide some comments and suggestions as follows. Hopefully, the following comments and suggestions will be helpful for improving this paper. 

Thank you for your valuable comments. We have tried to answer each comment below. A detailed description of the changes follows each comment.

=============

Major comments

---------------------

  1. This paper still lacks further explanation of the learning theory, especially the reference to the past research about the low education is a strong predictor of poor health and early death in adult age.

We have now added a few more text and references into the beginning of the introduction. These deal with the role of education in shaping health, and the mechanisms through which education may influence health.

  1. For the literature, it is recommended that the literature related to compare studies from other countries, in particular the Finnish database for this study, and to indicate the results of studies in other countries.  

We believe that the most relevant literature concerning chronic diseases and education choice or academic performance in adolescence is included. Most of that literature comes from other countries than Finland. In the data base of the present study, no similar articles have been done. Other studies using health registries have been done. We have added a reference to the analysis of register-based data from the Finnish 1987 birth cohort study (Ringbom et al. 2021). It is an example of how research has focused on mental health problems, and not on somatic chronic conditions. We are not sure, if we understood right the comment concerning “the Finnish database for this study”.

Minor comments

---------------------

  1. This study used the Linear regression analyses method. And 7th grade (N=8964) and 9th grade (N=7410) with 161 ending up to the non-academic upper secondary school. Is there any difference in the sample compared to other research? The other hand, Do the other control variables influence the result? Ex: family income? 

There must be some misunderstanding because the manuscript did not include “with 161 ending up to the non-academic upper secondary school”. This would, of course, be a false number. Our sample was a total sample of the students at the grades 7 and 9 in the Helsinki Metropolitan Area. Our data was based on the national registry of the applications to upper secondary schools and in practice all students apply via that system when completing the 9th grade. This means that the data is the best possible and represent the region. Compared to other parts of the country, those in the academic track is a bit higher in the Metropolitan Area. Our understanding is that such an application system does not exist elsewhere in the world or is anyway rare. We have changed some text in the methods section which clarifies the situation, hopefully.

The data did not include family income or corresponding socio-economic information, because children would not be able to answer questions regarding that. Parents’ education is known, e.g., PISA studies, to be a strong predictor of a child’s educational choice, most likely the strongest after school performance. And GPA is the main selection criterium for upper secondary studies, why it had to be taken as a control variable, too. This means that the most important control variables were included in the analyses. Because there could be comorbidities with chronic somatic disease and mental health problems, we added analyses when controlling for mental health problems in the paper. This did not change the results. 

5.The authors can discuss results, implications for managers and scope for future papers in different sections to enhance readability 

We made some changes in the manuscript to enhance readability. We also added a sentence of future research into the end of Discussion. 

This article still needs further explanation of the data, and the sample is not sufficient, it is recommended to make minor revise.

We have modified the description of the sample and the application registry in the methods sections and hope that these clarify the point. See also point 4 above.

Reviewer 2 Report

This is a valuable study. The data appears to be appropriate and the methodological choices are sound. The interpretations are reasonable and are well contextualised in the broader field. There are, however, a few issues that should be adressed.

In appendix A and B, the authors present estimates for the three conditions separately. Their in text-description is that the direction of association is similar to those found when modelling the conditions together. While this is of course true, my interpretation of the appendix tables A1 and A2 is that the results are in fact driven by a small group with epilepsy, with null och almost null findings for asthma and diabetes. Asthma and diabetes are by far more common than epilepsy so it appears plausible that this is not a power issue. While the authors description is true (the direction is the same) it would be more relevant to say thet the results are driven by epilepsy and focus the paper on that instead.

I miss a discussion on common co-morbidities of the included conditions. This could potentially give further insight into the mechanisms that generate the association. Common co-morbidities for epilepsy include depression and various learning disorders, for example.

On line 199- 200 the authors state that the study supports previous findings on chronic conditions and children with special needs. This gives ride to the question of how did the authors treat children with special needs. Are they included in the analysis? I suggest this group is excluded or included in a separate analysis since the mechanisms between chronic conditions and school performance is likely different in this group.

A descriptive table showing the distribution of the conditions by gender and parental education would be helpful. It would give the reader a sense of the data and how many are affected. I also suggest including the averages and prevalences of the outcome variables.

I find it hard to interpret the ORs from the logistic models. When the outcome is common (as it is appears to be here) ORs can be very large. As an alternative approach, I suggest fitting linear probability models, which would instead give the difference in percentage points. Since similar models are used (albeit with a continuous outcome) in the analysis of GPA it seems plausible that no major assumptions would be violated.

I found some inconsistencies in the reporting in the tables. In table 2 and 3, the grade categories are named “high grades”, “middle level grades” and “low grades”. I suggest a consistent naming strategy to avoid confusion. Furthermore, the number of value numbers reported are inconsistent in Table 1, 2 and A1. Please use a consistent number of decimals.

Author Response

Review 2

Open Review

English language and style

( ) Extensive editing of English language and style required
( ) Moderate English changes required
(x) English language and style are fine/minor spell check required
( ) I don't feel qualified to judge about the English language and style

Yes

Can be improved

Must be improved

Not applicable

Does the introduction provide sufficient background and include all relevant references?

(x)

( )

( )

( )

Are all the cited references relevant to the research?

(x)

( )

( )

( )

Is the research design appropriate?

( )

(x)

( )

( )

Are the methods adequately described?

(x)

( )

( )

( )

Are the results clearly presented?

( )

(x)

( )

( )

Are the conclusions supported by the results?

( )

(x)

( )

( )

Comments and Suggestions for Authors

This is a valuable study. The data appears to be appropriate and the methodological choices are sound. The interpretations are reasonable and are well contextualised in the broader field. There are, however, a few issues that should be adressed.

Thank you for your valuable comments. We have tried to follow the suggestions. A detailed description of the changes follows each comment.

In appendix A and B, the authors present estimates for the three conditions separately. Their in text-description is that the direction of association is similar to those found when modelling the conditions together. While this is of course true, my interpretation of the appendix tables A1 and A2 is that the results are in fact driven by a small group with epilepsy, with null och almost null findings for asthma and diabetes. Asthma and diabetes are by far more common than epilepsy so it appears plausible that this is not a power issue. While the authors description is true (the direction is the same) it would be more relevant to say thet the results are driven by epilepsy and focus the paper on that instead.

We have now added text into the end of Methods describing the Appendices A and B so that it better follows the findings.

I miss a discussion on common co-morbidities of the included conditions. This could potentially give further insight into the mechanisms that generate the association. Common co-morbidities for epilepsy include depression and various learning disorders, for example.

We adjusted the models for a variable indicating students’ self-reported mental health problem. We added text on this into the end of Results. This did not change the main results why we did not add any tables.

On line 199- 200 the authors state that the study supports previous findings on chronic conditions and children with special needs. This gives ride to the question of how did the authors treat children with special needs. Are they included in the analysis? I suggest this group is excluded or included in a separate analysis since the mechanisms between chronic conditions and school performance is likely different in this group.

In our data, there were altogether 20 answers from so called special schools. In Finland, children with special needs are mainly integrated in ordinary schools, sometimes in separate classes or groups. This information was not available. We did not exclude these because of the very small number.

A descriptive table showing the distribution of the conditions by gender and parental education would be helpful. It would give the reader a sense of the data and how many are affected. I also suggest including the averages and prevalences of the outcome variables.

This table has been added as the appendix table C and briefly opened in chapter 2.3, where the “parents’ education” variable is described.

I find it hard to interpret the ORs from the logistic models. When the outcome is common (as it is appears to be here) ORs can be very large. As an alternative approach, I suggest fitting linear probability models, which would instead give the difference in percentage points. Since similar models are used (albeit with a continuous outcome) in the analysis of GPA it seems plausible that no major assumptions would be violated.

ORs are a common way of presenting associations of categorised variables in public health. Changing the statistical modelling would not change the interpretation of the results.  

I found some inconsistencies in the reporting in the tables. In table 2 and 3, the grade categories are named “high grades”, “middle level grades” and “low grades”. I suggest a consistent naming strategy to avoid confusion. Furthermore, the number of value numbers reported are inconsistent in Table 1, 2 and A1. Please use a consistent number of decimals.

We have corrected the names of the categories and used a consistent number of decimals. However, we didn’t change the numbers of decimals, because we wanted to be more precise in reporting the small figures in the models for linear regression analysis.

Round 2

Reviewer 2 Report

I am happy to see that the manuscript have been improved in several ways. Still, the manuscript needs to be further revised before it is suitable for publication.

Major

Tables A1 and A2 show that the association between educational outcomes is strong for epilepsy but weak for asthma and null for diabetes. The coefficients are negative but given the crude measure of socioeconomic position of parents, and lack of other indicators of the child’s environment, it is difficult to conclude that the negative coefficients are not due to residual confounding from these types of factor. I understand that the authors may not share my interpretation, but this needs to be more carefully interpreted in the manuscript. The current interpretation of the results - that do not differentiate between the included conditions - is only partially supported by data. The methods section needs to be edited, acknowledging the weak/null associations found for asthma and diabetes. The  different results for the different conditions also need to be adressen in the discussion and preferrably in the conclusions and abstract.

The inclusion of mental health measures is an crucial improvement to the manuscript. The results from the model adjusting for mental health are the most convincing model. These estimates need to be shown, both to improve the credibility of the authors’ claims based on the results, and to more clearly position the study in the wider literature.

Thank you for clarifying that children with special needs cannot be distinguished. However, the 20 responses from the special schools should be excluded. The results are mainly driven by 80 children with epilepsy and if epilepsy correlates strongly with special needs this could potentially bias the results.

I do not find the convention of presenting OR:s in public health as a particularly convincing argument for using them here. In fact, the known issues of comparing OR:s obtained from different models is arguably making it more difficult to advance the field, since estimates from different studies, or even different models in the same study, cannot be easily compared. If the authors are not willing to fit LMP:s, at least present the equivalent AME:s to the OR:s in the supplement.

Minor

The second sentence in the introduction (line 34-35) appears to be missing some words at the end. Increase the risk [of what?].

Author Response

Manuscript ijerph-1947287: "A chronic disease in adolescence and the selection to an educational path – a longitudinal study".

Reviewer 2, Round 2

            Thank you for your further comments. Please find our answers below.

I am happy to see that the manuscript have been improved in several ways. Still, the manuscript needs to be further revised before it is suitable for publication.

Major

Tables A1 and A2 show that the association between educational outcomes is strong for epilepsy but weak for asthma and null for diabetes. The coefficients are negative but given the crude measure of socioeconomic position of parents, and lack of other indicators of the child’s environment, it is difficult to conclude that the negative coefficients are not due to residual confounding from these types of factor. I understand that the authors may not share my interpretation, but this needs to be more carefully interpreted in the manuscript. The current interpretation of the results - that do not differentiate between the included conditions - is only partially supported by data. The methods section needs to be edited, acknowledging the weak/null associations found for asthma and diabetes. The  different results for the different conditions also need to be adressen in the discussion and preferrably in the conclusions and abstract.

We have now more specifically stated that the results for epilepsy differ from those for the other two diseases. We have modified the sentences to show that the strongest associations with the outcome variables were the strongest for epilepsy.

We also have added text about epilepsy in Discussion, and mention it also in the Abstract. In Conclusions, we say that there is a need to do research on how single diseases and their comorbidities may affect educational outcomes, and that the life contexts of the adolescents should be taken into account in research.

The inclusion of mental health measures is an crucial improvement to the manuscript. The results from the model adjusting for mental health are the most convincing model. These estimates need to be shown, both to improve the credibility of the authors’ claims based on the results, and to more clearly position the study in the wider literature.

We have added a fourth model adjusting for mental health problem and the results did not change. We also added text explaining why this was included.

Thank you for clarifying that children with special needs cannot be distinguished. However, the 20 responses from the special schools should be excluded. The results are mainly driven by 80 children with epilepsy and if epilepsy correlates strongly with special needs this could potentially bias the results.

            We deleted the students from special schools (n=20). The results did not change.

I do not find the convention of presenting OR:s in public health as a particularly convincing argument for using them here. In fact, the known issues of comparing OR:s obtained from different models is arguably making it more difficult to advance the field, since estimates from different studies, or even different models in the same study, cannot be easily compared. If the authors are not willing to fit LMP:s, at least present the equivalent AME:s to the OR:s in the supplement.

It seems to vary by discipline whether Odds ratios (relational probabilities) or marginal effects (probabilities) are used. We use here odds ratios but added the marginal effects in Appendix D.

Minor

The second sentence in the introduction (line 34-35) appears to be missing some words at the end. Increase the risk [of what?].

            We have clarified this and say “… increase the risks for poor health outcomes.”